# Scan-Centric, Frequency-Based Method for Characterizing Peaks from Direct Injection Fourier Transform Mass Spectrometry Experiments

**DOI:** 10.3390/metabo12060515

**Published:** 2022-06-02

**Authors:** Robert M. Flight, Joshua M. Mitchell, Hunter N. B. Moseley

**Affiliations:** 1Markey Cancer Center, University of Kentucky, Lexington, KY 40536, USA; robert.flight@uky.edu (R.M.F.); jmmi243@uky.edu (J.M.M.); 2Department of Molecular & Cellular Biochemistry, University of Kentucky, Lexington, KY 40536, USA; 3Resource Center for Stable Isotope Resolved Metabolomics, University of Kentucky, Lexington, KY 40536, USA; 4Institute for Biomedical Informatics, University of Kentucky, Lexington, KY 40536, USA; 5Department of Toxicology and Cancer Biology, University of Kentucky, Lexington, KY 40536, USA

**Keywords:** Fourier transform mass spectrometry, orbitrap, scan-centric peak characterization, Fellgett’s advantage, interferograms, frequency spectrum

## Abstract

We present a novel, scan-centric method for characterizing peaks from direct injection multi-scan Fourier transform mass spectra of complex samples that utilizes frequency values derived directly from the spacing of raw *m*/*z* points in spectral scans. Our peak characterization method utilizes intensity-independent noise removal and normalization of scan-level data to provide a much better fit of relative intensity to natural abundance probabilities for low abundance isotopologues that are not present in all of the acquired scans. Moreover, our method calculates both peak- and scan-specific statistics incorporated within a series of quality control steps that are designed to robustly derive peak centers, intensities, and intensity ratios with their scan-level variances. These cross-scan characterized peaks are suitable for use in our previously published peak assignment methodology, Small Molecule Isotope Resolved Formula Enumeration (SMIRFE).

## 1. Introduction

Fourier transform mass spectrometry (FT-MS) provides high performance in terms of sensitivity, resolution, and mass accuracy all in one analytical instrumentation. The combination of these capabilities provides several analytical and interpretive improvements: (i) the ability to resolve distinct isotopologues with identical unit masses but different accurate masses [1]; (ii) enabling multi-element isotopic natural abundance correction for at least the lower portion of the detected mass range [2,3,4]; (iii) improved assignment accuracy [5,6]; and (iv) the detection of metabolites in the sub-femtomolar range, when combined with chromatographic separation [7,8]. In the metabolomics field, these improvements permit more complicated, but more informative experimental designs such as the use of single and multiple isotope-labeled precursors in stable isotope-resolved metabolomics (SIRM) experiments [9]. These stable isotope tracing experiments provide a wealth of isotope flux data that is interpretable in terms of metabolite flux information that is specific to a metabolic model, pathway, and subcellular location [1,10,11,12,13,14,15,16,17].

While these advantages of FT-MS are significant, when deployed in a high-throughput environment, the volume of data produced requires automated tools for data reduction, quality control, feature assignment, and downstream analyses. Furthermore, within the context of direct infusion, assignment of FT-MS spectral features lacks orthogonal sources of information such as chromatographic retention times, reducing the reliability of assignment with most MS assignment software tools.

Within this combined high-throughput with direct injection context, we previously developed methods to remove high-peak-density artifacts [18], annotate peaks with assignments using SMIRFE [6], and generate lipid classifications for spectral peak assignments [19]. We applied all of these methods together in detecting and assigning differential lipids in non-small-cell lung carcinoma (NSCLC) [20].

The input peak data in these studies are derived from multi-scan direct-injection FT-MS data. SMIRFE uses expected relationships of relative peak height or intensity to natural abundance probability (NAP) across multiple peak pairs to determine the likelihood of an assignment for the peaks in question. In the Thermo–Fisher Orbitrap FT-MS instruments, data are acquired in microscans and scans, where each scan is an aggregate of multiple microscans. While these microscan and subsequent scans are expected to be analytical replicates, spray instability and temporal delay in automatic gain control pragmatically break this expectation. Therefore, it is advantageous to keep the data at the scan level, so that any given poor quality (i.e., “bad”) scan can be removed prior to aggregating the data across scans and generating a centroided peak *m*/*z* and intensity. If the scan level data are not processed and aggregated correctly, the final peak intensities in arbitrary units and centers in *m*/*z* will have high uncertainty (total variance) [21].

In particular, if peaks are missing in some scans due to low abundance, the final aggregate peak intensities will not fit the expected relative intensity to NAP relationships. Here we describe novel scan-centric FT-MS metabolomics data processing methods that better preserve the expected relative intensity to NAP. In addition, due to the tendency of increasing point spacing with increasing *m*/*z* in Orbitrap type instruments, we derive a method for transforming the *m*/*z* data to axial frequency, which has the desirable property of being equally spaced across the full spectrum. Finally, we show that in addition to better preserving the expected intensity to NAP relationships, our scan-centric method also results in improved relative standard deviations, automatic identification of high-peak-density artifact peaks, and better separation of samples in a lipidomics data analysis.

## 2. Results

### 2.1. Simplistically Averaged Data Have Bad Relative Intensities

To motivate our solution, we generated peak lists using the peak exporting functionality in Xcalibur as well as our scan-centric peak characterization, and found matching Xcalibur peaks to our assignments. As an example, in Figure 1A, we show the Xcalibur calculated intensities from the simplistically averaged spectrum and theoretical peak intensities based on NAP for four peaks matching threonine. As the peak intensity decreases, the deviation between the Xcalibur calculated intensity and its corresponding NAP-based theoretical intensity becomes larger. This becomes more apparent when plotting the two sets of intensities directly against each other, as shown in Figure 1B.

### 2.2. m/z to Frequency

FT-MS data from the Thermo–Fisher Orbitrap instruments used to acquire the data do not provide any information about the raw spectral frequency data. Outside of the meta-data, it merely contains the *m*/*z* and intensity values for profile spectra acquired across multiple scans. However, the spectral frequency can be calculated by dividing the midpoint *m*/*z* of two adjacent points by their difference (Figure 2A, red points representing the midpoint *m*/*z* of two adjacent points, length of red lines representing the difference between the two adjacent points). The subsequent adjacent point differences in frequency are expected to be relatively constant with respect to *m*/*z*, as shown in Figure 2C–E, in contrast to the adjacent point differences in *m*/*z*, which are not constant with respect to *m*/*z*. The Thermo–Fisher Fusion instrument from which most of our collaborators’ data has been acquired, at a resolution of 450 K or 500 K depending on the sample, has an adjacent frequency point difference mode of 0.5, as shown in Figure 2D. It is important to note that although the frequency difference has been consistent regardless of instrument resolution, we do not know a priori what it should be, and the method makes no assumptions about the frequency difference, but rather calculates the mode of the frequency differences from those observed in the data itself. Restricting to those points that fall into a narrow range around the mode of frequency differences (0.49–0.51 in this work), a regression model of frequency to *m*/*z* can be generated (see Methods), with an example shown in Figure 2F. This regression model seems to fit the known relationship between frequency and *m*/*z*, where the frequency is related to $1/\sqrt{m/z}$. The actual regression model used in this work includes terms for *m*/*z*, $\sqrt{m/z}$, and $\sqrt[{3}]{m/z}$ (see Conversion of *m*/*z* to Frequency in Methods for further description of the regression model). We did investigate a variety of frequency regression models to see which one seemed to be the best, as shown in Appendix A.

The constant point difference in frequency space is useful, because some of the subsequent steps in our workflow use sliding and tiled windows where it is assumed that the sliding windows contain the same number of data points. The *m*/*z* point-to-point differences are not constant, but can be approximated by a LOESS linear model [22]; however, it is exceedingly difficult to create a LOESS model with an intercept of 0. In addition, we would also need to vary the width of sliding windows according to the *m*/*z* difference at a particular *m*/*z* based on the LOESS model. Frequency-based points suffer from none of these drawbacks, and the conversion from *m*/*z* can be derived from the raw profile level data itself, which is incredibly useful.

The *m*/*z* to frequency regression models are calculated for each scan, and the square-root term from all scan level models are checked for outliers based on the interquartile ranges across all scans in a sample. While scan specific models could be used in the conversion of *m*/*z* to frequency, doing so results in changes to the relative peak ordering compared to *m*/*z* space, as shown in Figure 3. Therefore, a single model for all scans based on the scan with the slope closest to the median of slopes across scans is used for converting *all* remaining scan level data.

Although the original model is created from only those points that had frequency point-to-point differences within a narrow range, all *m*/*z* points are converted to frequency for subsequent steps in the workflow.

### 2.3. Sliding Window Density to Remove Noise

In a dataset of this nature, we expect that much of the data are really just noise, and do not contribute meaningfully to the analysis. Furthermore, it is expected that noise is randomly distributed across the scans. Therefore, if we slide a window across the data and sum the number of non-zero points in each window, we expect that most of the data we encounter is actually noise. Subsequently, we divide the counts into tiled regions of a set size (1000 frequency in this work), and examine the 99th percentile of the sliding window counts within the tiled region. In Figure 4A, we plotted the non-zero point intensities from all scans for one tiled region 1000 frequency wide. The number of non-zero points in the sliding windows across this region are shown in Figure 4B, and zoomed closer in Figure 4C. These provide a measure of the non-zero point density across this region. The blue line indicates the cumulative 99th percentile for this one region, and is high, given that it looks like there are some consistent peaks across the scans in this region. However, examining the cumulative 99th percentiles across all the regions of the spectrum, as shown in the histogram of Figure 4F, a rather large fraction of the regions have very low cumulative 99th percentile values, with a median value of 5, which results in a cutoff value of 8 (ceiling of 5 × 1.5). Keeping only those sliding regions with a non-zero point count greater than 8 (above the red line in Figure 4B,C) results in those sliding regions shown in 4D, and then the point intensities in 4E. A non-zero point density-based method for removing potential noise is desirable because the peak intensities across scans are not identical, as shown by the need for normalization (see Normalization of Scans), and the distribution of RSD values before and after normalization (see Changes in Relative Standard Deviation).

Although it appears that only a handful of potential signal regions are removed between Figure 4A,E with a cutoff of 8 non-zero points in a sliding region, the number of initial signal regions rapidly explodes as the cumulative percentile cutoff is lowered, going from thousands to tens of thousands of potential peak regions to find and fit peaks within, as shown in Figure 5A,B for both the amino acid ECF and lipid samples, respectively.

### 2.4. Peak Characterization Using Quadratic Fit

Although many other types of mass spectrometry data suffer from a variable and noisy baseline, the scan-level profile data from the Thermo–Fisher Fusion have a baseline of 0 due to manipulations in the Thermo–Fisher firmware, making the determination of the centroided values considerably easier. For each region initially created, the peaks in each scan within that region can be characterized (i.e., centroiding). For centroiding, we use a simple weighted quadratic model of log-intensity to *m*/*z*. We add a small constant to enable using the zero values, and weight the values by their ratio to the most intense value, which is normally the value closest to the center of the peak, helping to ensure that the true centroid is determined. From the fitted model, we can derive the centroided center and the intensity of the peak, as shown in Figure 6.

### 2.5. Breaking Up Initial Regions

With the characterized (centroided) peak data from across scans within each region, it is then important to determine if only one or multiple “peaks” are actually present in the region. Our solution to this is to define breaks between actual peaks as a single frequency bin with zero characterized peaks within it. The frequency bins are created from tiled windows that are one frequency point difference wide. Adjacent non-zero frequency bins are merged to comprise a single peak region. Figure 7 shows an example where an initial region is broken up into two regions based on the characterized peak centers.

### 2.6. Normalization of Scans

Due to differences in how many ions are captured in the trap and the limited dynamic range of the detector, the observed peak intensities for the same analyte vary between scans (see Figure 7A,B for example). Using the median peak differences between scans, it is possible to normalize the peak intensities across scans. However, there are two issues with these peak intensities across scans: (a) some peak intensities correlate with the scan number (i.e., scan acquisition order); and (b) some peak differences between scans are correlated with intensity. The solution to **a** is to do a two-pass normalization. After the first pass, the peaks whose intensity is correlated with scan order are detected (absolute Pearson correlation > 0.5). In the second pass, the correlated peaks are removed, and normalization is carried out again. Figure 8A shows an example peak whose intensity across scans is correlated with scan number. The solution to **b** (peak differences correlated with intensity) is to only use the most intense peaks, as shown in Figure 8B. The highlighted peaks in Figure 8B are those with an intensity greater than 0.7 of the maximum intensity observed in that scan, and at least visually, their differences are not correlated with intensity. If *all* peaks are used for normalization, a very different set of normalization factors will be generated than by using only the *most intense* peaks, as shown in Figure 9A,B. Though the two-pass normalization (doublenorm) and single-pass normalization using the most intense peaks (singlenorm_int) generate very similar normalization factors (Figure 9B), this is expected as they both use the most intense peaks. However, removing the scan-correlated peaks does change the majority of the normalization factors.

### 2.7. Mitigation of High Peak Density Artifacts

We have previously described the presence of high peak density (HPD) artifacts in FT-MS spectra [18]. Ideally, the peak characterization procedure should reduce and mitigate their presence and effect in the resultant reported peaks. Their presence should be minimized by removing noise peaks and removing peak regions that have multiple reported peaks from the same scan. However, we expect they may also present as characterized peaks that have larger than expected frequency level standard deviations (FSD) when calculated across scans. These peaks can be detected by simply examining the distribution of FSDs and marking outlier peaks. To verify the mitigation of HPD regions, we converted centroided *m*/*z*’s from Xcalibur to frequency values using the previously calculated regression model for that sample, measured peak density to detect HPD regions, and compared them with the scan-centric peaks and FSD outliers. The Xcalibur peak density was measured using a sliding window ten points wide and a stride of one point.

Figure 10 shows a single HPD site detected in the 2ecf sample, with the peaks from Xcalibur, as well as various scan-centric processing and the peaks from centroiding using MSnbase. From this figure, we can see that the point-density-based noise filtering removes a large number of the peaks in the HPD site, while the FSD outliers removes further peaks that may be suspect. After removing the high FSD peaks, the number of peaks left in the HPD site are the same or less than those from MSnbase, and at least in this example, look more likely to be real peaks compared to those from MSnbase. Therefore, scan-centric characterization allows us to keep what are likely real peaks in HPD sites, without consideration of peak intensity, and mark peaks that may be artifactual from HPD regions.

### 2.8. Changes in Relative Standard Deviation (RSD)

Each step in the peak characterization either changes the overall number of peaks coming from each scan (sliding windows and breaking initial regions) or the overall intensity of the points within a scan. Therefore, one way to quantify any potential improvements in the characterized peaks is to look at the relative standard deviation (RSD) for the characterized scan level peak intensities (calculated as the standard deviation of peak heights across scans divided by the mean peak height) and compare them as each processing step is introduced. Figure 11 illustrates the peak height RSD distributions for four different samples. Up to two modes are reported for each distribution, where a mode location is only reported if it is ≥0.2 × the most intense mode. For samples 1ecf and 2ecf, there is a general shift in the RSD to the left, going from the bottom processing methods (msnbase_only) to the top processing method (filtersd), representing a visually clear improvement. For samples 49lipid and 97lipid, the improvements are visually more subtle, having a slight shift to the left as well as a narrowing and smoothing of the RSD distribution. The narrowing of the distribution is detected in the change of the standard deviations (SD) of the RSDs, however. Part of this visual subtlety is likely due to the bi-modality of these two distributions. This is not surprising, since these two samples are non-polar extractions from tissue and are biochemically more complex. Table 1 provides more quantitative metrics. At two decimal places, the filtersd, doublenorm, and singlenorm processing methods give superior and nearly identical results for three of the four samples, especially in terms of mean and median RSD. However, for sample 1ecf, the max RSD is much higher for singlenorm, highlighting its instability.

The 97lipid filtersd RSD distribution (as well as others) is bi-modal. Figure 12 plots each peak RSD as a function of the mean intensity across scans, for every peak present in at least three scans (top), or the peaks present in at least 80% of the scans (bottom). At least part of the bi-modality in the RSD distribution appears to be related to this dependence of RSD on intensity, as well as truncation as peaks appear in fewer scans. This is supported by part of the distribution disappearing when we require that peaks be in at least 80% of the available scans (difference between Figure 12 top and bottom).

We can remove a majority of the bi-modality by only examining those peaks with a Log10(mean) intensity ≥ 5. Appendix A show how the RSD distributions change for each sample when only the peaks with Log10(mean) intensity ≥ 5 are used. The distributions are all shifted to lower RSD; however, the overall trends in RSD are the same.

Figure 12 also provides information about the sources of variance or error in the FT-MS measurements. These trends or RSD with intensity (and SD with intensity, see Appendix A) imply both a constant, baseline additive error component that is independent of intensity, and a proportional error component where the variance increases with intensity. When the intensity is log-transformed, this additive component becomes a dispersive variance component. In addition, there is an unusual boot-shaped curve at the lower log mean intensities, representing left-censorship (truncation) effects due to detection limits.

### 2.9. Difference to Relative Natural Abundance

As an alternative to RSD, we can also compare the fit of relative intensities after assignment using SMIRFE [6] to the theoretical relative natural abundances (relNAP) of the assigned isotopic molecular formula’s (IMFs) within the assigned elemental molecular formulas (EMFs). Theoretically, we expect lower quality data to have both lower numbers of assignments, and for those things that are assigned, the fit between relative intensity and relNAP to be worse. To compare relative NAP to relative abundances, we only examined the assignments from the two samples containing ECF derivatized amino acids, as we can limit the assignments to those that match expected derivatizations of the known amino acids.

Figure 13 compares the peak-to-peak isotopic natural abundance probability and height log-ratio differences (Equation (4) in Methods) generated using heights from Xcalibur and from our scan-centric peak characterization. From this figure, it is clear that inconsistency in peak presence across scans leads to larger deviations between measured peak heights from aggregate spectra and expected relNAP.

Although we expected that the corrected scan-centric intensities would behave similarly or better compared to the raw intensities, Appendix A shows that while they have smaller differences than Xcalibur, there is still a trend of increasing difference with the raw scan-centric intensity—NAP ratios as the number of missing scans increases.

### 2.10. Method Specific Peaks

Each set of peaks generated may or may not be specific to the particular method used to generate centroided peak *m*/*z* and intensities. This is true for the different scan-centric combinations, as well as the peaks from Xcalibur and MSnbase. Here we examine the overlap of the unassigned (Figure 14) and assigned peaks from the full scan-centric processing with Xcalibur and MSnbase. These same counts are also summarized in Table 2 and Table 3. In these two examples, there are some striking differences. The 1ecf sample has all of the scan-centric peaks shared with either Xcalibur or MSnbase peaks, whereas the 97lipid sample has 2/3 of the peaks specific to scan-centric characterization and not matched to either of the other methods. Notably, for both samples, the scan-centric characterization produces similar numbers of peaks, even though the upper mass limit in 1ecf is 1000 *m*/*z* compared to 1600 *m*/*z* for the 97lipid sample, whereas the number of peaks from MSnbase and Xcalibur are three-fold and 40-fold higher in the 1ecf sample compared to the 97lipid sample.

Figure 15 shows the distribution of differences between the observed and expected *m*/*z* of assigned peaks in the 1ecf and 97lipid spectra. For the 1ecf-specific histograms, it is clear that these differences have a narrower unimodal distribution from the scan-centric peak characterization, especially in comparison to MSnbase. In addition, the MSnbase distribution has far fewer peaks that match a scan-level assigned peak. Given the large number of MSnbase-characterized peaks present in the spectrum, many peaks may be outside the matching tolerance of 2 ppm. This strongly implies that the peak center error is far higher in the MSnbase-characterized peaks than what the histogram directly shows. For the 97lipid-specific histograms, again the difference distribution for the scan-centric peak characterization is narrower, but the improvement is not as pronounced, likely due to the distribution being bi-modal. Moreover, far fewer Xcalibur and MSnbase characterized peaks matched assigned scan-centric peaks. Again, this strongly implies that their peak centers have far higher error. When we matched assigned peaks across the NSCLC samples for examination of changes in *p*-values (see Changes in *p*-Values on a Large Dataset), the standard deviation of the peak locations in frequency space was 0.5 (results not shown).

### 2.11. Changes in p-Values on a Large Dataset

For large datasets, we expect that more correct peak intensities will result in better agreement between sample normalized peak intensities within a sample class. One way to test this is to compare *p*-values generated from the different intensities. Figure 16 compares the −1 × Log10(*p*-values) from 373 (corrected and Xcalibur) or 87 (MSnbase) isotope resolved molecular formulas (IMFs) assigned by SMIRFE for the scan-centric peaks and then peaks matched and intensities extracted from the other methods. Although all of the *p*-values are somewhat (and statistically) different from those reported by the raw intensities, there are some interesting patterns of differences. MSnbase generated *p*-values show the widest distribution of differences, as well as the smallest number of peaks that are present in 50% of both the cancer and non-cancer samples. This echoes the patterns of low overlapping assigned peaks observed in both ECF samples, where many of the scan-centric amino acid peaks only found one match from the MSnbase peaks. Surprisingly, the truncated log-normal distribution corrected intensities generated very different *p*-values compared to the raw intensities, and Xcalibur showed the most agreement with the raw *p*-values. We tested the statistical significance of these differences using a *t*-test of the actual difference in log-*p*-values (shown in Figure 16D). It is clear that the raw scan-centric intensities are superior.

It is vitally important, however, to normalize the intensities correctly. Appendix A shows that with the incorrect normalization, the corrected and raw *p*-values become much closer to each other.

### 2.12. Quality Control and Quality Analysis

Having a scan-centric workflow for generating the peak centroids means that we have opportunities to evaluate the scans as a whole, as well as the peaks across scans. For example, we can mark peaks with unusually high frequency standard deviations (FSD) and mitigate the presence of high-peak-density artifacts (see Mitigation of High Peak Density Artifacts). We also sometimes find that from the initial peak region that has been marked as a single region, it still results in two peaks being characterized in a single scan. This results in that peak being removed from that scan entirely.

The conversion of *m*/*z* to frequency also provides opportunities to evaluate each scan and whether it should be kept for further processing. Checks that have helped us in the development of the scan-centric peak characterization have included: (a) verifying that the ordering of data points remains identical in *m*/*z* and frequency space after conversion to frequency; and (b) examining the R^2^ fit of the predicted frequency points to the original frequency points across scans for outliers. A third quality control step is to examine the coefficients of the square-root term for each scan (see term **y** in Equation (1)) and look for any outliers, and remove them before continuing the sample processing. Across the 169 NSCLC raw files that completed scan-centric processing, 105 had at least one scan removed based on the square-root coefficient. Figure 17A shows a histogram of the number of scans removed based on being outliers of the overall distribution. Figure 17B shows what outlier scan coefficients look like in a single sample.

A final check we implemented in the conversion to frequency was verifying that the range of frequency point-to-point differences allowed for creating the *m*/*z* to frequency model are the same across scans. The initial set of scans we started with included all of the MS1 scans, including the precursor scans for the MS2 scans. However, the samples were acquired as a mix of MS1 only scans for the first 7.5 min, and then a mix of MS1 precursor scans followed by MS2 scans for the second 7.5 min. In 20 samples, the MS1 precursor scans have a different resolution than the MS1 main scans, as shown in Figure 18. This is easy to detect by differences in the square-root coefficients, but due to using a single frequency model for converting all scans actually manifested as different modes in the point-to-point frequency differences. Oddly enough, the check for outlier scans discussed previously, does not flag any of the scans as outliers in this case. Ultimately, we were able to resolve this anomaly by only using those scans with an injection time before 7.5 min. In the future, the package will be updated to check that the resolution is the same across the scans, and produce an informative error for the end user before quitting.

Finally, in the scan-to-scan normalization step, we require that there be at least 25 peaks in each of the scans that can be used to calculate normalization factors between the scans and the reference scan. If there are no scans with at least 25 peaks, then the processing of that sample will error and stop.

## 3. Discussion

FT-MS direct-injection data provide spectral scans; however, unlike nuclear magnetic resonance, these spectral scans are not identical, due to several instrumentation issues. Therefore, simple scan summation or averaging to an aggregate spectrum followed by peak characterization does not provide optimal data. Additionally, this approach prevents the calculation of useful peak-specific statistics, especially variances, both in *m*/*z* and intensity. In the results presented, we demonstrate superior performance by scan-centric peak characterization in terms of: (a) improved QC/QA that removes low quality scans; (b) intensity-independent noise reduction that pragmatically eliminates HPD sites; (c) improved peak height relative standard deviations; (d) improved peak intensity ratios that better match natural abundance probabilities; and (e) better separation between biological groups. These improvements are necessary for downstream derivation of molecular formula by SMIRFE using spectrum-derived tolerances, an efficient *m*/*z* search of a large isotope-resolved cache, and filtering by peak ratio matching to NAP. However, scan-centric peak characterization requires a sophisticated pipeline of QC/QA and processing steps to derive a superior characterization with informative statistics and QC/QA metrics. Years of methods development, testing, and optimization have gone into the methods presented here. Nonetheless, once implemented, large datasets can be processed in a straightforward manner that supports full computational reproducibility. We also kept all steps that provided a demonstrable improvement, no matter how small that improvement was, but the following steps provided the most improvement: (a) noise removal by non-zero point density, (b) scan normalization using median peak intensity differences, and (c) removal of outlier scans. However, both steps a and c require the *m*/*z* to frequency regression model. In the end, it is the combination of steps that synergistically provide the presented superior peak characterization results. We note that the non-zero point density-based noise removal is particularly useful, as the peak intensities from scan to scan are not identical, which would possibly hamper any kind of intensity-based noise removal. 

Not all of the methods presented here provided an improvement. We included our negative results with correcting mean and standard deviations for the intensity of peaks not detected in all scans, so that others do not waste effort pursuing this. We initially expected the data to follow a truncated log-normal distribution, since Figure 12 clearly demonstrates truncation effects; however, upon further contemplation of the results presented here, we believe that the differences between scans still present after normalization imparts a dispersive variance component. Moreover, Figure 12 (and Appendix A) demonstrates the presence of an additive error component that will become dispersive after log-transformation. Thus, we hypothesize that a truncated negative binomial distribution may better represent the intensity of peaks not detected across all scans. Fortunately, this distribution has been previously studied [23] and we plan to explore this possibility for a future improvement to our methods. However, correction may not be straightforward and it is unclear if it is better to apply the correction to the raw or log-transformed peak height data. Either way, a dispersive variance component is present and must be accounted for. As illustrated in Figure 12, there are low intensity peaks that are observed across the majority of the scans. This likely indicates that the loss of peak detection involves other factors that are increased at low intensity. We theorize that any factor that increases signal decoherence, such as ion-cloud repulsion, would increase the peak flooring performed by the instrument. Therefore, access to the scan level peak intensity information as provided by the methods presented here facilitates the inference of sources and types of error present in the measurements, which we believe have been previously under-described and is required for future data processing improvements.

## 4. Materials and Methods

### 4.1. Samples and Overall Processing

Two different sets of samples were used to evaluate the various methods: ethylchloroformate (ECF) derived amino acid samples described in [6] and non-small-cell-lung-carcinoma (NSCLC) lipid-extracted tissue samples described in [20].

The method for generating the amino acid samples was adapted from a previously published method for performing ECF amino acid derivatization [9]. Two replicate samples were prepared, and spectra were obtained for both samples using a Tribrid Fusion Orbitrap at 500 k resolution and a mass range of 150 to 1000 *m*/*z*.

The collection, preparation, and mass spectrometry analysis of the paired cancer and non-cancer samples has been previously described [14]. In summary, cancer and nearby non-cancer tissue samples were acquired from eight-six non-diabetic patients with suspected resectable stage I or IIA non-small cell lung cancer. Written informed consent was collected from all subjects prior to inclusion and all samples were collected under a University of Louisville or University of Kentucky IRB protocol. Lipid extracts were prepared using a modified Folch extraction and reconstituted for direct infusion ultra-high resolution mass spectrometry on a pair of Thermo–Fisher Tribrid Fusion Orbitrap instruments coupled to an Advion nanoelectrospray system. Two of the cancer lipid samples (49Cpos and 97Cpos) were used for the majority of examples in this manuscript. The rest were included for the examination of changes in differential analysis *p*-values using different peak source intensities (see Differential Analysis of Large Dataset).

For each of the two ECF and two lipid samples, the raw data file was converted to profile mzML format. Only the MS1 scans were used, and the scan-to-scan time difference had to be ≥4 s, and for the lipid samples, only scans acquired before 450 s (7.5 min) were kept. The scan-centric data were then processed in the following eight ways:No noise removal, no normalization (noperc_nonorm);Noise removal, no normalization (perc99_nonorm);Noise removal, single-pass normalization with all peaks (singlenorm);Noise removal, single-pass normalization with high ratio peaks (singlenorm_int);Noise removal, two-pass normalization (doublenorm);Noise removal, two-pass normalization (filtersd);Scans merged and then centroids generated by MSnbase (using combineSpectra and pickPeaks);Scans merged and peak-list exported by Xcalibur.

### 4.2. Matching Peaks

To associate the Xcalibur and MSnbase peaks with the scan-centric peaks, a 4 ppm (2 ppm low and high) window is calculated for the scan-centric centroid, and if any peaks are found within the window, the one with the smallest *m*/*z* difference is kept as the matching peak to the scan-centric one under consideration.

### 4.3. Conversion of m/z to Frequency

The input data consist of profile mode *m*/*z* spectra from multiple scans encoded as *m*/*z* and intensity values for each scan. No information about the original observed frequency values is available in either the raw files or the mzML files. However, proxy frequency values can be generated by averaging the *m*/*z* of adjacent points and dividing them by the *m*/*z* difference. Ideally, the difference between subsequent points in this proxy frequency space is constant, in practice there is a range of differences in frequency space. The actual, constant difference can be obtained by examining the median of the calculated frequency differences, and then constraining *useful* points (those that can be used for generating a model of frequency to *m*/*z*) to be within 2% of the mode value. These *useful* points can be used to construct a linear model relating *m*/*z* to frequency for individual scans based on the formula:(1)$$frequency=a+\frac{x}{mz}+\frac{y}{\sqrt{mz}}+\frac{z}{\sqrt[{3}]{mz}}$$

From the known physical properties of the Orbitrap, only the square-root term should be necessary [24]. Practically, we found the combination of no root, square and cube-roots to provide a better fit when processing data from the Thermo–Fisher Fusion instrument, likely due to issues with slight imperfections in the orbitrap geometry, contributions from space charge effects and magnetronic motion, control of the magnetic fields, and the Fourier-like transform conversion used by the spectrometer. We do note that when working with Bruker SolariX ICR data, the equation is slightly different, and does not require the cubic square-root term (unpublished results, see Appendix A). A frequency model was generated for each scan, followed by a single model using the scan with the square-root term closest to the median of the square-root terms from all scans. We observed that this single model better preserved the relative ordering of the peaks in both *m*/*z* and frequency-space compared to the scan specific models (see Results).

To convert *m*/*z* back into frequency, we can use a similar model without the roots, as well as an extra simple linear term that does not have an equivalent in the above frequency model.
(2)$$\begin{array}{cc} mz=a+x\times \frac{1}{frequency}+y\times \frac{1}{frequency^{2}}+z\times \frac{1}{frequency^{3}} & \end{array}$$

### 4.4. Frequency Intervals

Two types of frequency intervals were used: sliding and tiled windows. In this work, the sliding windows are the equivalent of 10 frequency points wide with a stride of one point. The tiled windows are one point wide with a stride of one point. Each point above is the equivalent of the difference between data points in frequency space, which in these samples have a spacing of 0.5.

### 4.5. Interval Range Based Data

To enable interval algebra, the frequency points were converted to single width intervals by multiplying by a constant factor to maintain the differences in individual points (a multiplier of 400 in this work), rounding to the nearest integer, and storing them as IRanges objects from the IRanges Bioconductor package [25]. The sliding and tiled windows were also converted to IRanges objects using this process.

### 4.6. Peak Containing Intervals

To find intervals that contain points that represent actual signal and not just random noise, the number of non-zero intensity points in each sliding window were counted. Subsequently, we broke these counts into fixed width tiles (default width of 1000 in frequency space) and calculated the cumulative 99th percentile of non-zero points for each tile. The rounded up median value × 1.5 of these 99th percentile values from the fixed width tiles were used as the cutoff value to determine which of the initial sliding regions should be kept as regions containing potential signal. Any sliding window with a non-zero count less than or equal to the cutoff value is removed, and the remaining sliding windows are kept and overlapping sliding windows are merged to create the initial peak regions. The presence of zero intensity points in these Thermo–Fisher Orbitrap spectra are primarily due to flooring implemented by the spectrometer when local spectral quality falls below a certain threshold. When this flooring is unstable and incomplete, a partial ringing phenomenon is observed [18].

Within each initial interval region, peaks in each scan are detected (see Peak Detection and Centroided Values), and their centers are binned by the tiled windows. Adjacent tiled windows with non-zero peak counts are merged together, and any zero peak count tiled windows split the initial region into multiple peak interval regions. These interval regions should contain a single real peak that was detected in one or more scans.

### 4.7. Peak Detection and Centroided Values

On a single scan level, possible peaks are detected by simple bump-hunting for two increasing points followed by two decreasing points using the find_peaks function in the pracma package [26]. These possible peaks are then characterized using a weighted parabolic fit of log-intensity to position (where position is either *m*/*z* or frequency), and the weights for each point are the relative log-intensity compared to the maximum log-intensity for the peak.
(3)$$\begin{array}{cc} \ln\left(intensity\right)=a+x\times position+y\times position^{2} & \end{array}$$

From this weighted parabolic fit, the center, intensity, integrated area, and sum-of-square residuals can be extracted for the peak. The center and intensity from this model are equivalent to the centroided peak center and intensity.

Before further processing, each region is verified to have only one peak from each scan. If a scan has two or more peaks in a region, then the scan level data in that region is discarded. Any regions that subsequently contain zero peaks are removed.

### 4.8. Scan to Scan Normalization

Scans are normalized to a single *reference* scan based on the log-intensity differences in a subset of peaks present in at least the same number of scans as the 95th percentile of scan counts for the peaks. In addition, only those peaks with an intensity greater than 0.7 times the highest intensity peak in the scan are used. Pairwise scan-to-scan distances are calculated by taking the cartesian distances between log peak intensities present in both scans, and then the cartesian distance is summed across the scan-to-scan distance to provide an overall difference in each scan to all other scans. The scan with the lowest summed overall distance is chosen as the *reference* scan ($scan_{ref}$), and normalization factors for each scan are calculated as the median log peak intensity differences in $scan_{i}$ compared to the $scan_{ref}$. This normalization was carried out twice, once using all possible peaks, after which the correlation of peak intensity with scan order was checked, and those peaks with correlation of greater than 0.5 with scan order were removed, and the normalization factors were calculated again, and then applied to both the centroided peak height and the raw point intensities. Peaks correlated to scan order represent an artifact that we speculate results from a gradient in the sample well and are marked in the final output.

### 4.9. Full Scan-Centric Characterization

The full set of raw data points for each peak in each scan within a region is known based on the previously detected peaks. Therefore, the non-zero intensity, normalized raw data points across scans can be combined, and then characterized again using the weighted parabolic fit method previously described. In addition to the data from the full set of raw points, the means and the standard deviations of the peak height and location can be derived from the scan-level peak characteristics previously calculated.

In addition to these values, the frequency point-to-point median difference was calculated across all of the raw data points for those points that could be used for modeling frequency to *m*/*z*, and this difference of a single point from the peak center is calculated in frequency space, and converted to *m*/*z* space to provide an “offset” value that is potentially useful to define the search space around the peak for any assignment algorithm.

### 4.10. Correction of Height and Standard Deviation

Ideally, each peak would be observed in every scan. However, some peaks are not observed in some scans due to the number of ions falling below the detection threshold, or being excluded after filling the trap. This should result in a left-censored log-normal distribution of peak intensities across equivalent scans, which is expected for analytical measurements with detection limits [27]. To correct these, either a correction based on a model of the truncated normal distribution can be used on log-transformed data, or the differences can be simulated by sampling from data that is present in most of the scans. To simulate the effect of peaks missing from some scans on the standard deviations, the peaks present in all scans were used. For each peak, a sample of the heights across scans was obtained (ranging from 5% to 95% of scans), and a new standard deviation was calculated for that fraction, and a ratio of the fractional standard deviation to the “true” standard deviation was calculated. The ratio standard deviation across peaks can then be fitted to a cubic model of the fraction used, and a correction factor predicted for those peaks that are present in fewer scans. Our correction uses log-transformed peak heights, and differences instead of raw heights and ratios directly to make some of the calculations easier. The corrected standard deviations can then be used to correct the mean height assuming that it is the result of a left-censored normal distribution [28,29].

### 4.11. Marking High Frequency Standard Deviation Peaks

High-peak-density (HPD) artifacts [18] present as groups of singular peaks with higher-than-expected frequency standard deviations (FSDs) calculated from the scan-to-scan frequency peak locations. Outliers are detected by calculating the interquartile range (IQR) of the distribution of FSDs across the entire spectrum, and peaks with FSDs greater than the median plus 1.5 times the IQR (as implemented in boxplot.stats) are marked. The HPD detection algorithm from Mitchell et al. [18] was re-implemented in R for this work to allow comparisons between it and the use of the FSD. For HPD detection, the peaks in excel output from Xcalibur were used after converting the *m*/*z* peak centers into frequency space. Sliding windows that are 1000 frequency points wide with a stride of 100 points were used for the density calculations.

### 4.12. Calculation of Relative Standard Deviation

For consistency across both the various scan-centric centroiding and MSnbase, the scan level Log10 heights were converted to raw heights, and then averages and standard deviations calculated across scans, removing missing values. Relative standard deviation (RSD) was calculated as standard deviation divided by the mean for each peak. The number of scans a peak was present in was also noted, as well as the percentage of total scans. Only peaks present in at least three scans were kept for analysis of the RSD.

Modes of the RSD were calculated starting from a density approximation of the distribution using default settings in the density function of the R stats package. Peaks from the density approximation were determined with the find_peaks function in the pracma package. Up to two modes are reported from low to high RSD along the distribution: (i) the most intense peak; (ii) the next most intense peak if it is ≥0.2 × the most intense peak.

### 4.13. Scan-Centric Peak Assignment

Our previously described SMIRFE algorithm [6] was used to assign molecular formulas to scan-centric characterized peaks in an untargeted manner. For the lipid samples, an initial EMF database was generated using an *m*/*z* limit of 1605 *m*/*z*, and maximum numbers for each element were set to C: 130, N: 7, O: 28, P: 3, H: 230. Assigned formulas were allowed to have K^+^, Na^+^, H^+^, and NH_4_^+^ adducts (only positive mode samples were assigned). Assigned molecular formulas were then classified into one or more lipid categories using our lipid classifier tool [19]. For the ECF derivatized amino acid samples, the EMF database was generated using an *m*/*z* limit of 1005 *m*/*z*, and maximum numbers for each element were set to C: 100, N: 7, O: 40, H: 230, P: 3, S: 3. ^15^N labeling was included in the database generation. Assigned formulas were allowed to have H^+^ and Na^+^ adducts.

We attempted to create peaks that could be assigned by SMIRFE from the MSnbase and Xcalibur peak lists, but SMIRFE would not assign them given its requirements for scan-centric peak characterization data that includes variance.

### 4.14. Consistently Assigned Lipid Spectral Feature (Corresponded Peak) Generation and Peak Intensity Normalization

Our SMIRFE assignment method assigns isotope-resolved molecular formulas (IMFs) to characterized peaks in each spectrum. Each IMF represents an isotopologue of a given elemental molecular formula (EMF) (e.g., 13C112C51H1216O6 is an IMF representing the m+13C1 isotopologue of EMF C6H12O6). Consistently assigned formulas (i.e., corresponded peaks) were identified using an in-house method we have named EMF voting.

Elemental molecular formula (EMF) voting was used to match peaks across samples and determine the most likely assignments from SMIRFE. First, for each sample, the assignments were extracted, filtered, and scored. Any assignments that contained only peaks that were marked as being questionable (high FSD or correlated with scan number) were removed. Scores are calculated as 1 − E-value, and for the lipids, EMFs that were classified as lipids had their score multiplied by 2.

For each sample, peaks were grouped to a sample-specific EMF by determining the list of shared EMFs across a group of peaks (grouped_EMF). Scores for each formula in the grouped_EMF in the sample were obtained as the best score for that formula from available scores in the group. After all sample grouped_EMFs are generated, additional scores for a formula are considered in actual voting by looking for the same unadducted base EMF with different adducts and adding these scores into the final total score.

The pseudo_EMFs are collections of grouped_EMFs across samples. They were generated across samples by iteratively merging grouped_EMFs with shared formulas, creating a new list of formulas in an EMF, and merging any pseudo_EMFs that have shared formulas again.

For each pseudo_EMFs, the most likely formulas are determined by voting. Voting uses the sum of EMF scores across samples, including those from the same formula with a different adduct. Those formulas with a total score in the top 90% of all total scores were considered “winning” formulas and kept as the “voted formula” set. Any peaks that did not originally have a “voted formula” were then checked to see if the peak location was within previously defined tolerance, and if the ordered peak intensities for the set of peaks from a sample are in the same order as the natural abundance probabilities of IMFs for the voted EMF. If so, then the peaks are kept as having the “voted formula” set.

After voting, all of the peaks were checked to make sure that the peak locations are within previously defined *m*/*z* specific cutoffs. Any peak outside of its specific cutoff were removed.

Finally, those pseudo_EMFs that share greater than 50% of their peaks are merged together, and voting on the EMFs is performed again.

Peak locations from our custom data processing pipeline are reported both in *m*/*z* and frequency, which are derived from the *m*/*z* values. The frequency values are more reliable (i.e., more consistent with less variance), but differ between instruments. Therefore, for each set of samples from a particular instrument, we use high confidence assignments (e-value ≤ 0.1, *m*/*z* ≤ 600) to derive a frequency cutoff using the mode of the distribution of frequency standard deviations across both groups of samples. This frequency cutoff is used to do EMF voting within an instrument. The *m*/*z* standard deviations of the voted peaks are then fit to *m*/*z* using a generalized additive model, and the standard deviation of the predicted *m*/*z* values are multiplied by 2 to derive an *m*/*z* cutoff so that voting can be performed across the instruments.

EMF voting identified 3529 total corresponded peaks across all 165 spectra. All lipid isotopologue intensities were normalized by dividing the isotopologue intensity by the median intensity of all the peaks in the sample.

### 4.15. Peak—Peak NAP Height Ratios

Each pair of peaks in a specifically labeled and adducted elemental molecular formula are related by their natural abundance probability (NAP). Theoretically, the log-ratios of two peaks NAPs are approximately equal to the log-ratios of two peaks intensities.
(4)$$\begin{array}{cc} \ln\left(NAP_{P1}/NAP_{P2}\right)-\ln\left(Int_{P1}/Int_{P2}\right)\approx 0 & \end{array}$$

SMIRFE assignments include the NAP for each of the isotopologue molecular formulas (IMFs) in the particular labeled and adducted EMF. For each peak pair in the peak group for a particular EMF, we calculated the log-ratio for the NAPs and the log-ratio for the intensities, and their absolute differences. This calculation was carried out for the final characterized intensity, as well as the intensities at the scan level, and for both the *raw* intensity and *corrected* intensity, and for matched Xcalibur peak intensities, and matched MSnbase peak intensities if there were two or more matching peaks in the labeled, adducted EMF.

### 4.16. Differential Analysis of Large Dataset

To compare *p*-value changes in a large multi-class sample dataset, we used the full set of NSCLC lipid samples described earlier. We started with the 181 matched non-cancer and cancer samples previously used for HPD detection and lipidomics of non-small-cell-lung-cancer (NSCLC) [18,20]. All samples were characterized using the full scan-centric workflow, assigned using SMIRFE (see Scan-Centric Peak Assignments), and then peaks matched by shared EMFs across samples (see Consistently Assigned Lipid Spectral Feature (Corresponded Peak) Generation and Peak Intensity Normalization). After extraction of the scan-centric IMF peak locations across all samples, Xcalibur and MSnbase peaks were matched within each sample, and locations and intensities extracted.

Of the starting 181 samples, 12 did not finish during scan-centric peak characterization for various reasons, leaving 169. In addition, some of the samples were acquired multiple times, creating duplicate samples. Not knowing which sample run was most appropriate to use, we removed all of the duplicated samples. Using information-content-informed Kendall-tau correlation (ICI-Kt) [27], we compared the median correlations of each sample to all others in the same disease class for outliers, and removed 5 cancer and 5 non-cancer samples. The median ICI-Kt correlations are shown in Appendix A.

Each sample intensity was normalized by dividing by the median intensities from that sample and intensity method. Only those peaks present in 50% of both the non-cancer and cancer samples were kept for differential analysis. This resulted in 373 IMF peaks for differential analysis. Differential analysis used the logged intensity values, with *p*-values calculated using rowttests function from the genefilter Bioconductor package (v1.76.0) [30], removing any missing values before calculation. *p*-values were adjusted using the Benjamini–Hochberg method in the p.adjust function from the base R stats package (v4.1.0) [31].

For comparisons across peak sources, we converted both the raw *p*-values to log-*p*-values by calculating −1 × Log10(*p*-value). The reference peak *p*-value is the raw scan-centric *p*-values.

### 4.17. Software Used

Thermo–Fisher Xcalibur was used to export peak lists from all samples used. SMIRFE v 1.0 [6] running under Python 3.8 [32] was used for assignments. Lipid classifications were generated by LipidClassifier v 1.0 [19]. All other calculations were performed in R v 4.1.0 [31]. The targets package v 0.10.0 was used to control processing and aggregation of results [33], and renv v 0.15.3 [34] to create a reproducible R package environment. Plots were generated using ggplot2 v 3.3.5 [35], patchwork v 1.1.1 [36], ComplexHeatmap v 2.10.0 [37], ggridges v 0.5.3 [38], and ggforce v 0.3.3 [39]. MSnbase v 2.20.4 [40,41] provided facilities for reading in scan-level data, merging scans, and calculating centroided peaks. ICI-Kt correlation values among lipid samples were calculated using ICIKendallTau v 0.1.16 [27]. Outlier lipid samples were determined using visualizationQualityControl v 0.4.7 [42]. Specific data manipulation facilities were provided by dplyr v 1.0.8 [43], tidyr v 1.2.0 [44], furrr v 0.2.3 [45]. This manuscript was generated from rmarkdown v 2.11 [46,47,48].

## Figures and Tables

**Figure 1 metabolites-12-00515-f001:**
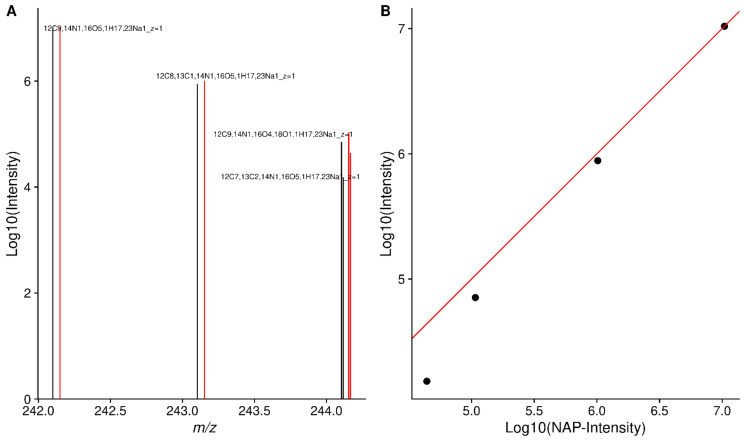
(**A**) Xcalibur intensities (black) and theoretical intensities based on relative NAP (red) for ECF derived threonine. The 18O isotopologue is shifted by 0.01 *m*/*z* and the NAP peaks by 0.05 *m*/*z* to aid visualization. (**B**) Theoretical NAP-based-Log10-intensities and Log10 Xcalibur intensities. The red line indicates perfect agreement.

**Figure 2 metabolites-12-00515-f002:**
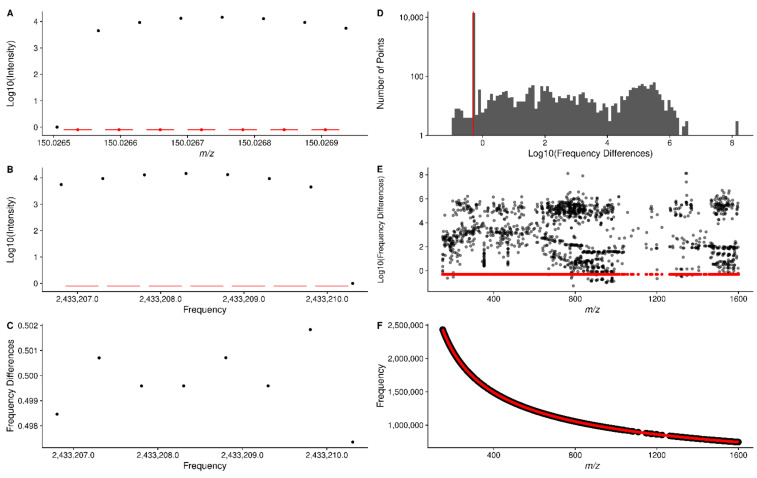
(**A**) Intensity vs. *m*/*z* for a single peak from a single scan. Red lines denote the difference between each adjacent pair of points, and red dots the *m*/*z* midpoint between the pair of points. The difference divided by the midpoint is used to derive the spectral frequency values in (**B**). (**B**) plots the intensity vs. the converted frequency points derived from (**A**). The red lines denote the difference between adjacent points, which are shown in the y-axis of (**C**) for this single peak. (**D**) shows a histogram of adjacent point differences for all points. Note that the y-axis is Log10 scaled. The differences for all points in a single scan vs. *m*/*z* are shown in (**E**), with those differences that lie within 0.49–0.51 shown in red. (**F**) shows the plot of derived spectral frequency vs. *m*/*z*, with fitted values from the linear regression model in red.

**Figure 3 metabolites-12-00515-f003:**
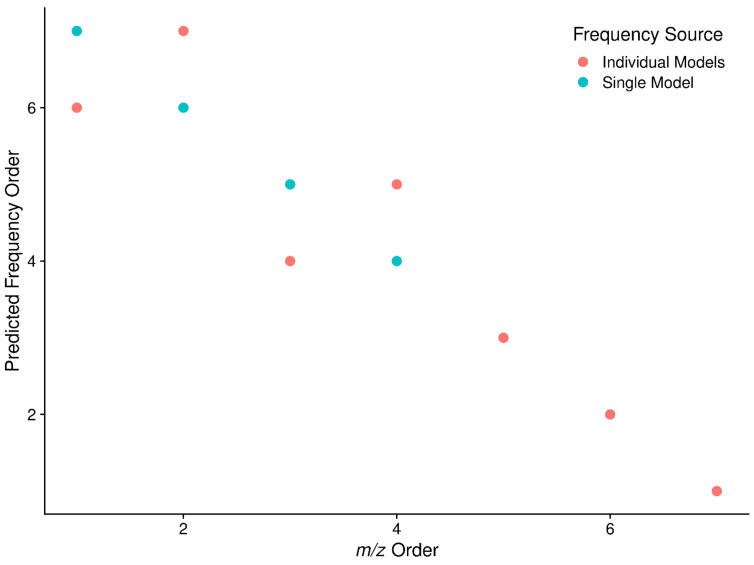
Peak ordering in *m*/*z* compared with ordering in frequency space when a single *m*/*z* to frequency model is used or scan specific *m*/*z* to frequency models are used. For a single peak, the scan level peak *m*/*z*’s were extracted, and then frequency values for those *m*/*z* generated using a single common model of *m*/*z* to frequency (*Single Model*), or models derived from each scan (*Individual Models*). A subset of the peaks ends up out of order using scan specific models, implying that a single model should be used across all the scan level data.

**Figure 4 metabolites-12-00515-f004:**
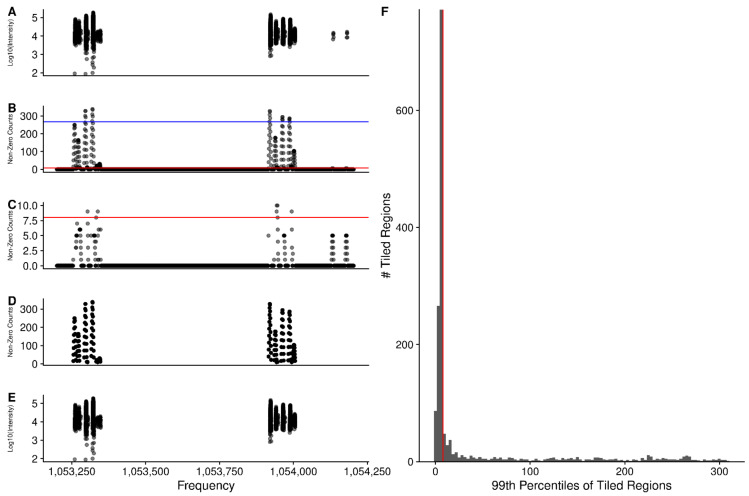
(**A**–**E**) A single tiled region 1000 frequency wide. (**A**) Non-zero point intensities in frequency space across all scans. (**B**) Number of non-zero points in sliding regions. The blue line indicates 99th percentile of non-zero counts in this one region. The red line is the rounded value of median × 1.5 of 99th percentiles from (**F**). (**C**) Zoomed y-axis of (**B**) to show the non-zero counts that are below the median cutoff. (**D**) Sliding region non-zero counts that are greater than the cutoff. (**E**) Non-zero point intensities across scans in the regions from (**D**). (**F**) Histogram of the 99th percentile cutoffs from all of the tiled windows. The red line is the median × 1.5.

**Figure 5 metabolites-12-00515-f005:**
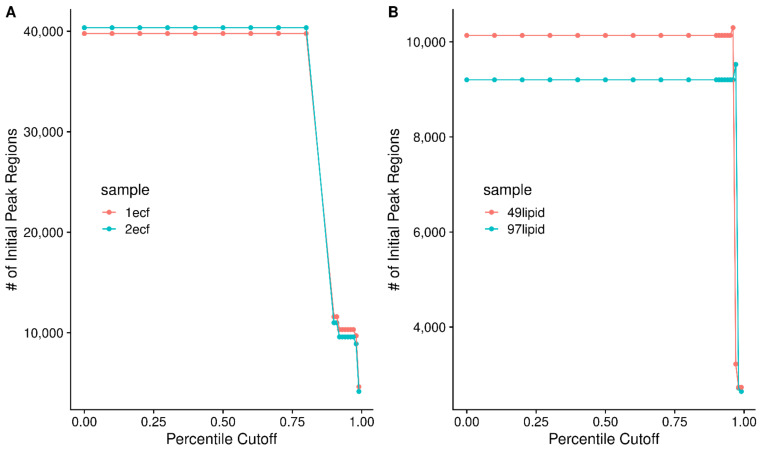
The number of initial regions as a function of the percentile cutoff used for either the AA ECF (**A**) or lipid (**B**) samples.

**Figure 6 metabolites-12-00515-f006:**
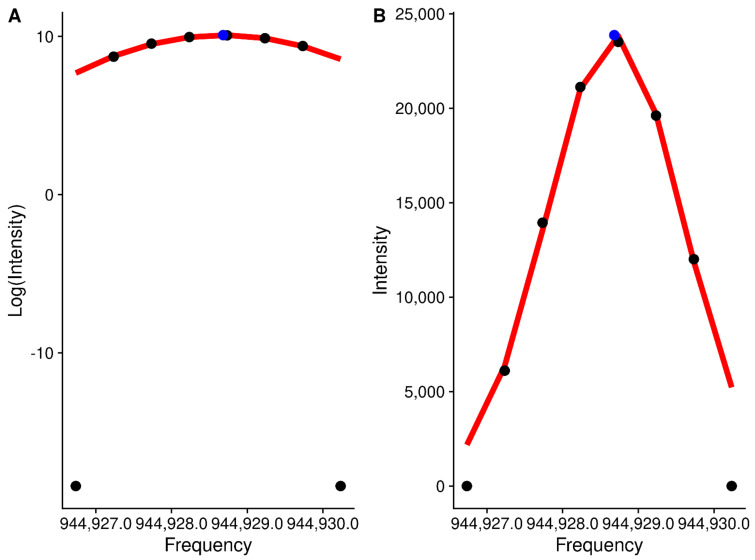
Log-intensity (**A**) and intensity (**B**) of points for a single peak against frequency. Black points are the original data points, and the blue point represents the calculated centroid.

**Figure 7 metabolites-12-00515-f007:**
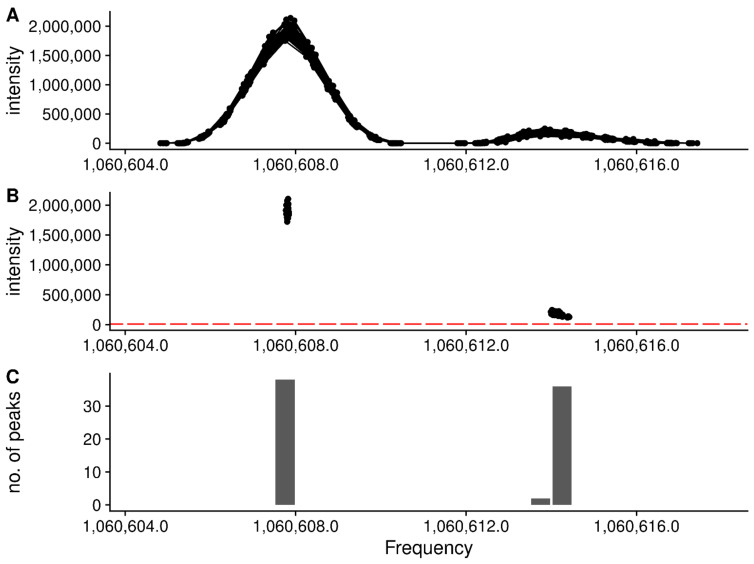
Splitting a single region into two regions based on the peaks that are present. (**A**) The full set of raw frequency and intensity data across all scans for the region are shown. Each horizontal trace are the point intensities in a single scan. Clearly the region has two separate peaks within it. (**B**) The peak centroids (frequency and intensity) for each peak are in black. The tiled regions (red) are used to quantify the number of peaks. (**C**) The number of peaks within each tiled region are shown as a histogram. Each group of non-zero adjacent regions will be merged to form a new peak region.

**Figure 8 metabolites-12-00515-f008:**
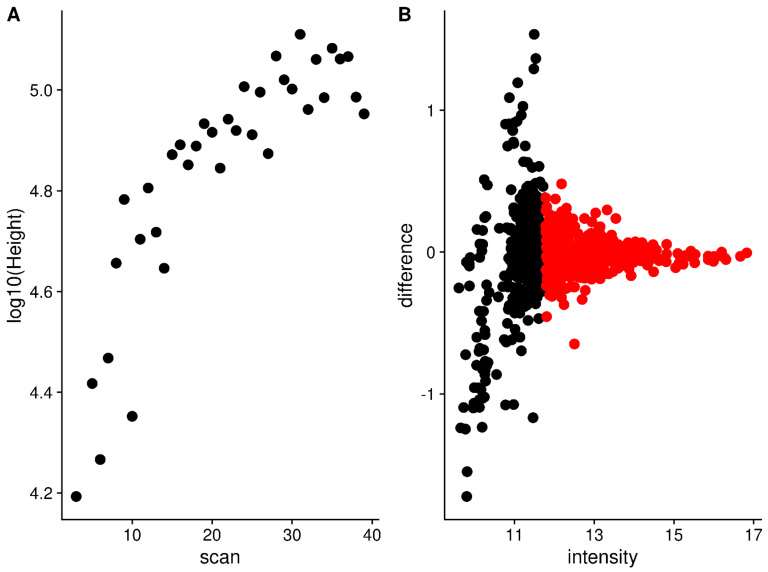
(**A**) An example of a peak whose height across scans is correlated with scan number. (**B**) The peak differences to the same peaks in a reference scan are plotted against peak height. Black: Peaks with a height < 0.7 of the maximum. Red: Peaks with a height ≥ 0.7 of the maximum.

**Figure 9 metabolites-12-00515-f009:**
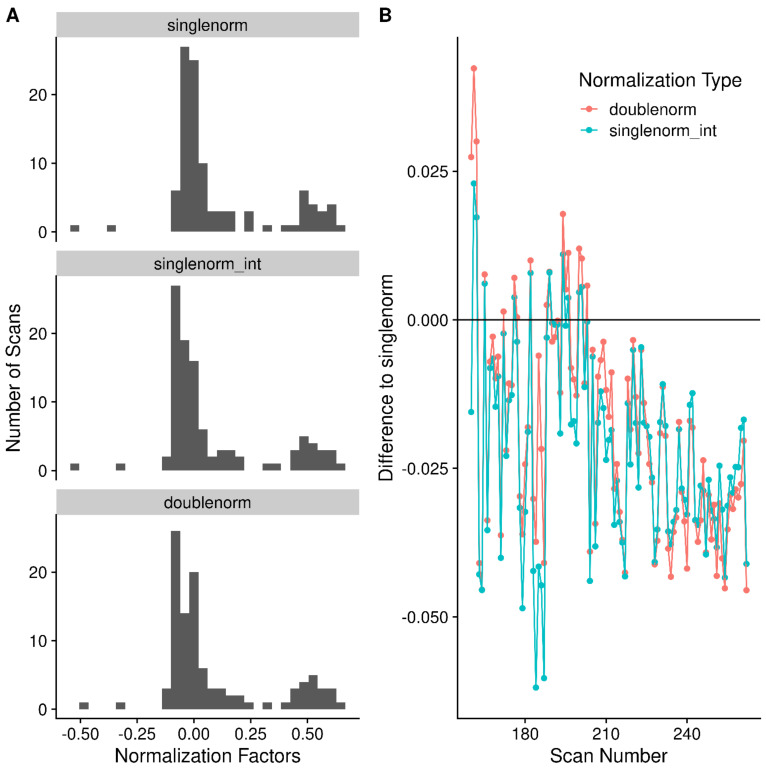
(**A**) Histogram of scan normalization factors using either a single-pass normalization using *all* peaks (*singlenorm*), single-pass normalization using peaks with an intensity ≥ 0.7 of the maximum intensity (*singlenorm_int*), or the two-pass normalization removing peaks whose height is correlated with the scan and using the most intense peaks (*doublenorm*). (**B**) The difference in the normalization factors obtained from either *doublenorm* or *singlenorm_int* compared to *singlenorm*.

**Figure 10 metabolites-12-00515-f010:**
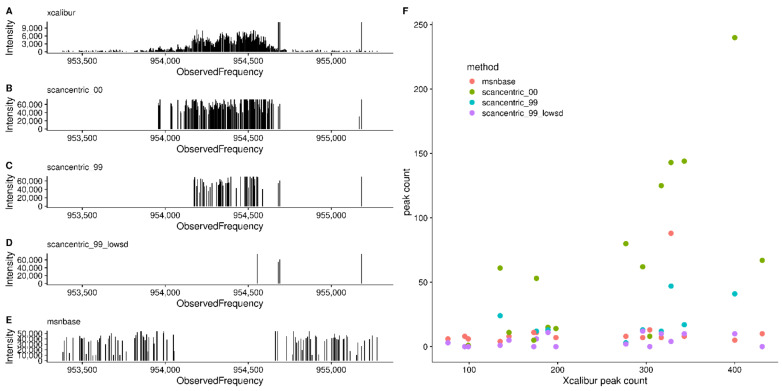
Comparison of HPD and high FSD sites in the 2ecf sample. (**A**–**E**) Peak plots for various peak processing modalities in a single HPD site. (**A**) *xcalibur*: peaks exported from Thermo–Fisher Xcalibur after averaging scans. (**B**) *scancentric_00*: Scan-centric peak characterization without any density-based filtering (see noperc_nonorm in Methods). (**C**) *scancentric_99*: Scan-centric peak characterization using the default point density filtering (see filtersd in Methods). (**D**) *scancentric_99_lowsd*: Same as *scancentric_99*, but removing any peaks marked as having a high frequency standard deviation. (**E**) *msnbase*: peak centroids generated from *MSnbase*. (**F**) Scatterplot of peak counts across all of the HPDs detected in the 2ecf sample against the Xcalibur peak counts.

**Figure 11 metabolites-12-00515-f011:**
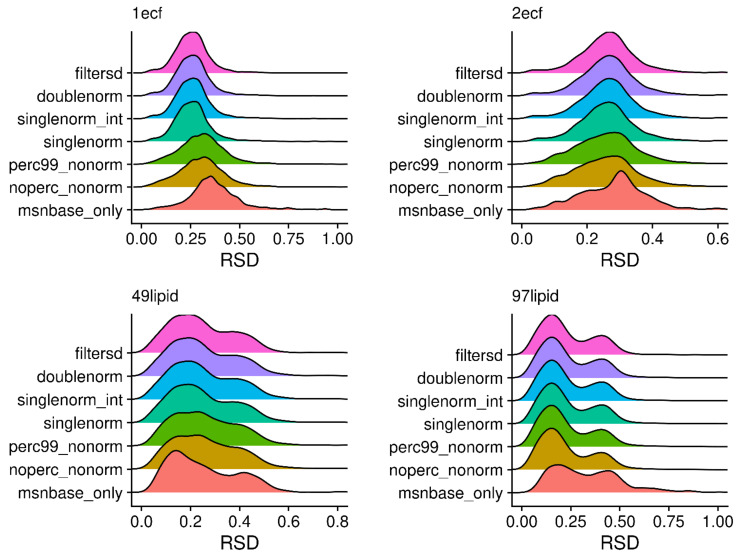
Density plots for relative standard deviations (RSD) of peak heights across scans for each of the processing methods. A peak was required to be present in at least three scans for the RSD value to be reported.

**Figure 12 metabolites-12-00515-f012:**
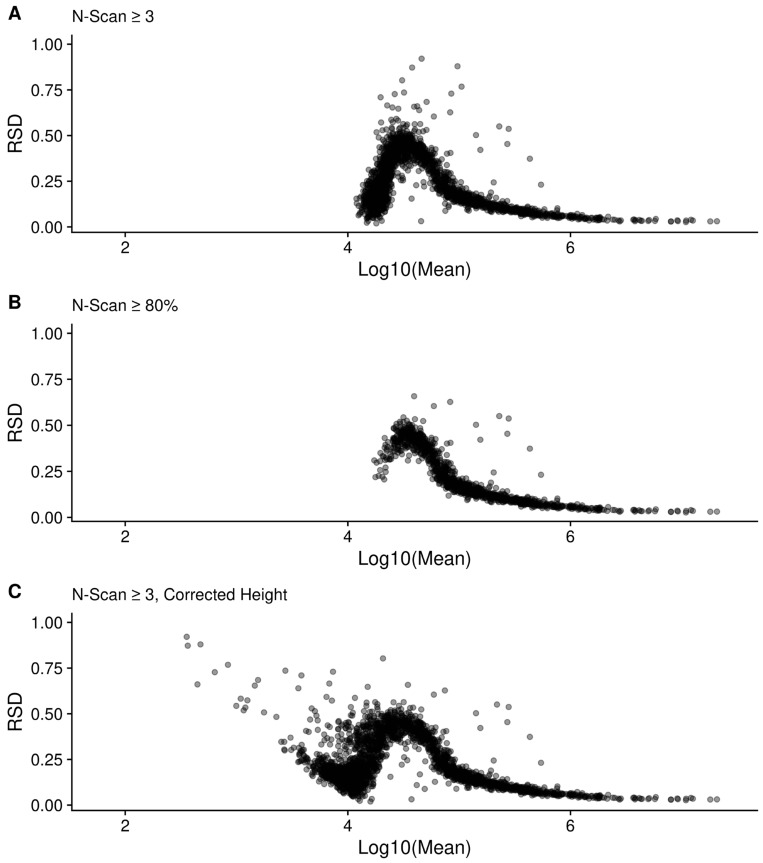
RSD as a function of the log-mean peak intensity for the filtersd method peaks from the 97lipid sample. (**A**) shows all peaks in at least three scans using raw height; (**B**) shows peaks that were present in at least 80% of the scans; and (**C**) shows all peaks in at least three scans using the corrected height.

**Figure 13 metabolites-12-00515-f013:**
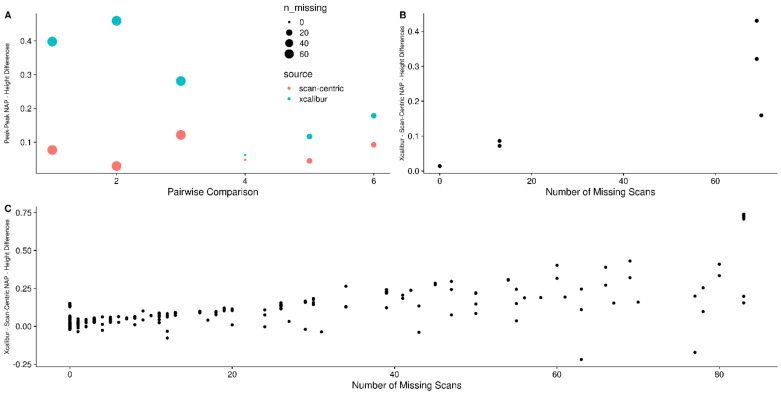
(**A**) The peak-to-peak NAP—intensity log differences from scan-centric peak heights (red) and Xcalibur peak heights (blue) from the ECF derivatized threonine amino acid assignments with Na adduct, with point size reflecting how many peaks were missing across scans. (**B**) The difference in Xcalibur to scan-centric ratios plotted directly as a function of the number of scans the peak was not found in. (**C**) The differences in Xcalibur to scan-centric ratios for all of the amino acid assignments in EMFs with more than a single peak in both ECF samples.

**Figure 14 metabolites-12-00515-f014:**
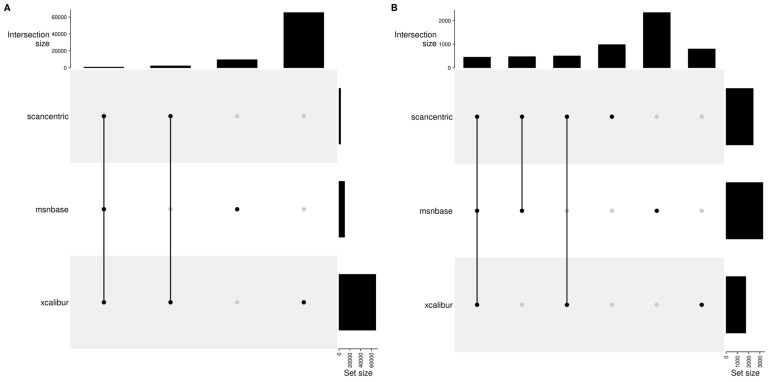
UpSet plot with the counts of common and specific peaks without consideration of assignments for each of scan-centric, MSnbase, and Xcalibur generated peaks for the 1ecf (**A**) and 97lipid (**B**) samples. The black points with connected vertical lines identify which set intersections are represented in the bar at the top. A single black point identifies what was specific to the set and not in any other sets.

**Figure 15 metabolites-12-00515-f015:**
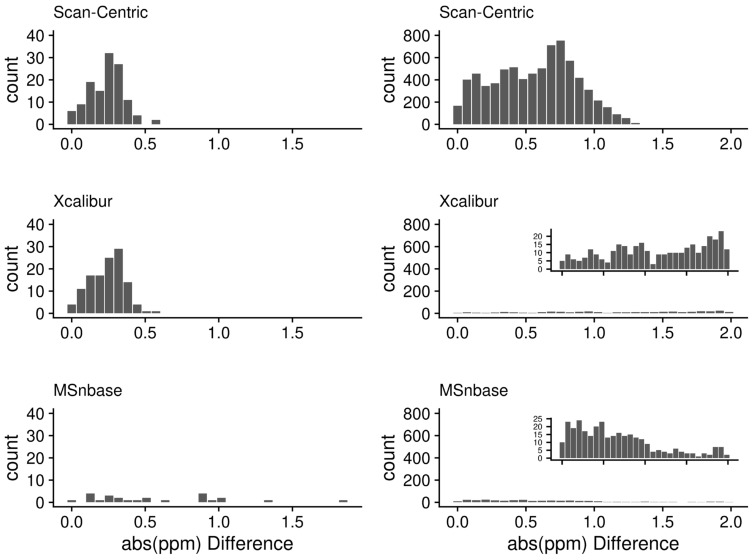
Histograms of the differences between observed and expected *m*/*z* values for assigned peaks in the 1ecf (**left**) and 97lipid (**right**) spectra, measured in parts-per-million (ppm) respectively. Inset plots are scaled to the maximum counts observed for the particular method.

**Figure 16 metabolites-12-00515-f016:**
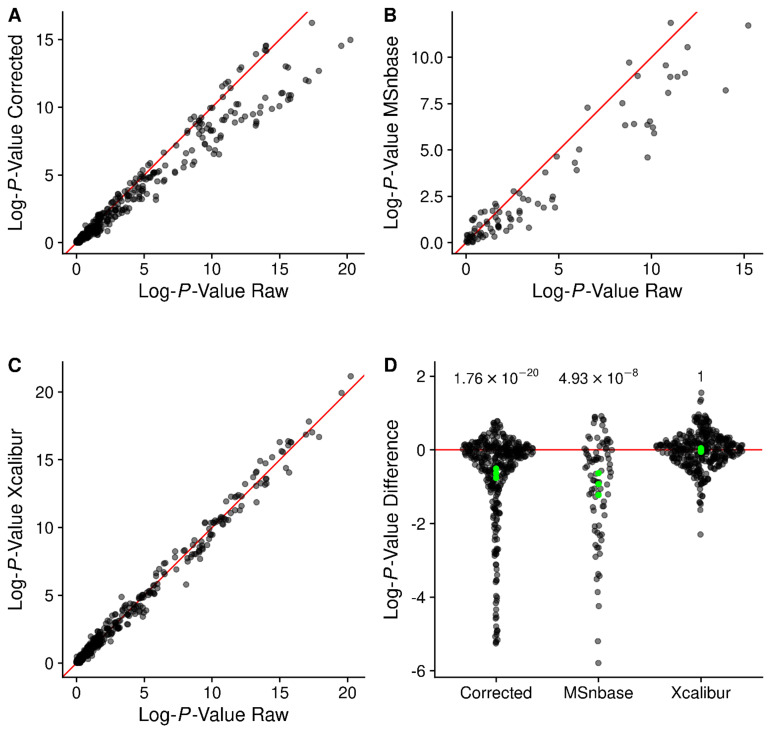
(**A**–**C**) Log-*p*-values generated by comparing non-cancer and cancer sample IMFs using peak intensities from different methods. Red line denotes perfect agreement. (**D**) Sina plot of differences in the log-*p*-values generated by different methods compared to the raw scan-centric log-*p*-values. The Bonferroni adjusted *p*-values from a *t*-test of the log-*p*-value differences for each method are also shown. Green points denote the high, mean, and low-confidence limits reported from the *t*-test.

**Figure 17 metabolites-12-00515-f017:**
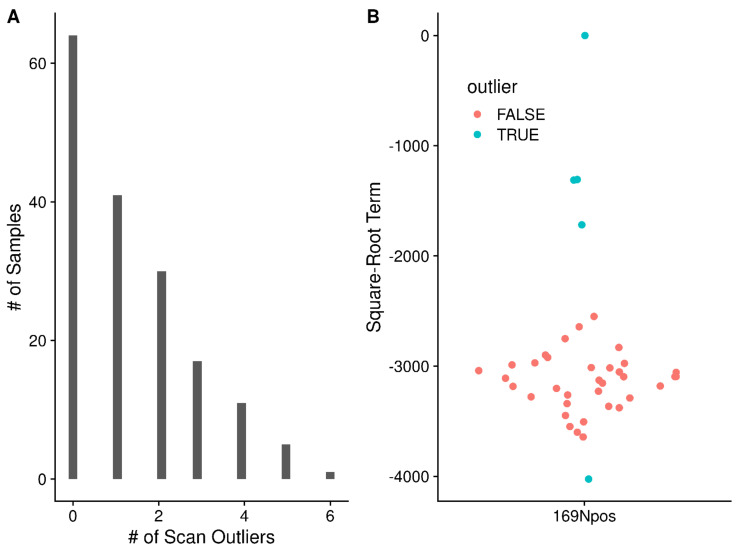
(**A**) Number of scan outliers for each sample in the NSCLC dataset. (**B**) Single example for sample 199Cpos of the scan square-root terms, with outliers marked and removed from further consideration.

**Figure 18 metabolites-12-00515-f018:**
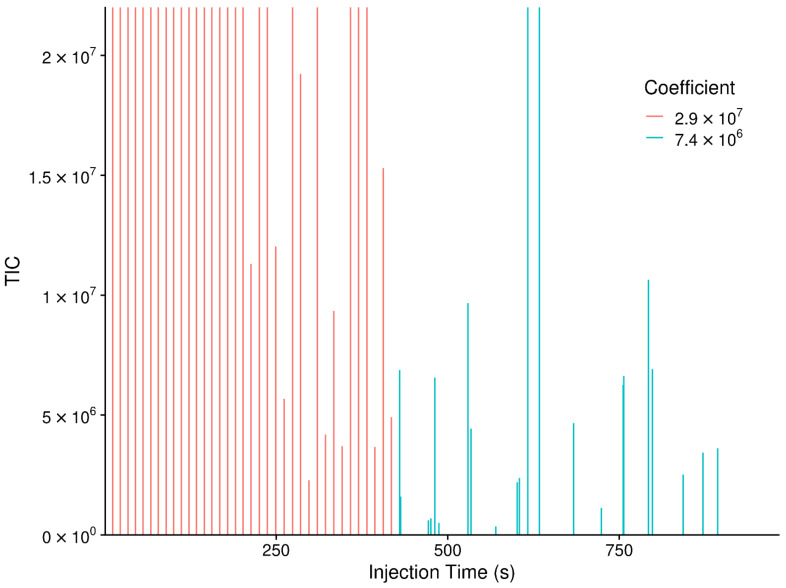
Scan-level total ion chromatogram (TIC) plot noting the square-root coefficient for each scan, where the coefficient is reflective of the resolution used for each scan. The y-axis is cropped to 2 × 10^7^ so that the MS1 precursor scans are visible, as their TIC’s are much smaller than the others.

**Table 1 metabolites-12-00515-t001:** RSD means, standard deviations (sd), medians, modes, and maximum observed values for each sample with different overall processing.

Sample	Processed	Mean	Sd	Median	Mode 1	Mode 2	Max
1ecf	filtersd	0.26	0.09	0.25	0.26		1.37
1ecf	doublenorm	0.26	0.09	0.26	0.26		1.37
1ecf	singlenorm_int	0.26	0.10	0.26	0.26		1.41
1ecf	singlenorm	0.26	0.10	0.25	0.28		1.43
1ecf	perc99_nonorm	0.31	0.12	0.31	0.32		1.19
1ecf	noperc_nonorm	0.31	0.12	0.30	0.32		1.19
1ecf	msnbase_only	0.37	0.14	0.36	0.35		1.19
2ecf	filtersd	0.26	0.09	0.26	0.27		1.01
2ecf	doublenorm	0.27	0.10	0.26	0.27		1.05
2ecf	singlenorm_int	0.27	0.10	0.26	0.27		1.05
2ecf	singlenorm	0.26	0.10	0.26	0.27		1.03
2ecf	perc99_nonorm	0.26	0.11	0.26	0.28		1.08
2ecf	noperc_nonorm	0.26	0.11	0.26	0.29		1.08
2ecf	msnbase_only	0.29	0.11	0.29	0.30		0.99
49lipid	filtersd	0.25	0.13	0.23	0.20	0.37	1.13
49lipid	doublenorm	0.25	0.14	0.23	0.19	0.37	1.13
49lipid	singlenorm_int	0.25	0.14	0.23	0.20	0.37	1.13
49lipid	singlenorm	0.25	0.14	0.23	0.19	0.37	1.13
49lipid	perc99_nonorm	0.27	0.15	0.25	0.16	0.23	1.13
49lipid	noperc_nonorm	0.27	0.15	0.25	0.16	0.23	1.13
49lipid	msnbase_only	0.26	0.15	0.22	0.14	0.42	1.14
97lipid	filtersd	0.24	0.14	0.20	0.16	0.41	2.05
97lipid	doublenorm	0.24	0.15	0.20	0.15	0.41	2.05
97lipid	singlenorm_int	0.24	0.15	0.20	0.16	0.41	2.05
97lipid	singlenorm	0.24	0.15	0.20	0.15	0.41	2.04
97lipid	perc99_nonorm	0.23	0.14	0.19	0.15	0.41	2.03
97lipid	noperc_nonorm	0.23	0.14	0.19	0.15	0.41	2.03
97lipid	msnbase_only	0.34	0.20	0.30	0.19	0.43	1.94

**Table 2 metabolites-12-00515-t002:** Number of matched peaks between peak processing methods for 1ecf. There are no overlapping peaks only between scan-centric and MSnbase, as well as no peaks found only by the scan-centric method.

Method					Set_Sizes
Scan-centric	x	x			2937
Xcalibur	x	x	x		68,244
MSnbase	x			x	10,330
comb_sizes	778	2159	65,307	9552	

**Table 3 metabolites-12-00515-t003:** Number of matched peaks between peak processing methods for 97lipid.

Method							Set_Sizes
Scan-centric	x	x	x	x			2405
Xcalibur	x	x			x		1747
MSnbase	x		x			x	3263
comb_sizes	448	502	472	983	797	2343	

## Data Availability

The code used in the reporting of results is available on GitHub at https://github.com/MoseleyBioinformaticsLab/manuscript.peakCharacterization (accessed on 1 April 2022), and archived on Zenodo [49]. The scan-centric peak-characterization is available as an R package at https://github.com/MoseleyBioinformaticsLab/FTMS.peakCharacterization (accessed on 1 April 2022), and the specific version used in this work is archived on Zenodo [50].

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
