# Peer review of "Scan-Centric, Frequency-Based Method for Characterizing Peaks from Direct Injection Fourier Transform Mass Spectrometry Experiments"

_metabolites, 2022, doi:10.3390/metabo12060515_

Round 1

Reviewer 1 Report

Mass spectrometry-based metabolite identification and characterization are rapidly developing. Ultra-sensitive detection, high throughput analysis, and intelligent data processing strategies are enabling the identification of very low abundant metabolites and other biomolecules. Downstream identification of metabolites is based on the FT-MS signal in mass spectra. The well-characterized peaks lead to confident identification. In the current manuscript, the authors reported the methods to characterize the peaks to improve the identification. The authors are quickly describing the different methods without much of its discussion. However, the manuscript is interesting and could be further improved.

Major points:

  1. In Results section 2.1. the authors are trying to demonstrate that the Xcalibur calculated intensities and theoretical peak intensities differ with the help of Figure 1. The observed and theoretical intensities often differ and it is well-known fact. What new things that authors are trying to show are hard to find. Figure 1 is hard to understand and there is not enough description of that. 
  2. “The Thermo-Fisher Fusion instrument from which most of our collaborators data has been acquired, at a resolution of 450K or 500K depending on the sample, has a mode of 0.5, as shown in Figure 2C and D” is not clear as if the frequency difference range already within 0.5 or should be within that range? Also, the peak ordering calculations from m/z to frequency models need more description.
  1. In the section “Sliding Window Density to Remove Noise” the authors describe the method to remove the noise. The authors need to give a real example demonstrating the removal of noise from the actual signal scans. Not all no-zero points are noise. This needs to be better explained as it is not clear how the authors reached a cutoff of 7.5.
  2. Breaking up the peaks when the co-isolated ions are overlapped may be helpful. The authors may want to clearly demonstrate the advantages of splitting the peaks of non-zero values.
  3. Normalization of Scans, Mitigation of High Peak Density Artifacts, and Changes in Relative Standard Deviation (RSD) need more discussion. The singlenorm_int and the two-pass normalization are expected to be similar due to the most intense peak selection.  
  4. The authors indicated that not all methods reported in the manuscript are improved, and they also reported negative results. This is highly valuable in science. However, the authors need to include which methods are really improved and which are not. This would be very helpful to draw conclusions.

Minor points:

  1. Abstract needs revision.
  2. Figure 1 could be improved. From where does the 18O isotopologue appear?
  3. Figure 13 is hard to understand and could be improved. 

Reviewer 2 Report

Please see the attached

Reviewer 3 Report

The text suggests an important statistical approach to improve the data. More chemometric analyzes would be needed to improve data processing. good job

Round 2

Reviewer 1 Report

The authors satisfactorily addressed all the major and minor points and provided enough description in the results and discussion. The manuscript is improved after revision and can be accepted.